# The Effects of Emergency Room Violence toward Nurse′s Intention to Leave—Resilience as a Mediator

**DOI:** 10.3390/healthcare9050507

**Published:** 2021-04-28

**Authors:** Jui-Hsuan Li, Ta-Wei Chen, Hsiu-Fang Lee, Whei-Mei Shih

**Affiliations:** 1Department of Nursing, Chang Gung Memorial Hospital, Linkou 333423, Taiwan; lovevarl@cgmh.org.tw (J.-H.L.); f22066@cgmh.org.tw (H.-F.L.); 2Attending physician of Emergency Medicine, Chang Gung Memorial Hospital, Linkou 333423, Taiwan; sputnik615@cgmh.org.tw; 3Department of Nursing, Chang Gung University of Science & Technology, Taoyuan 33324, Taiwan; 4Graduate Institute of Gerontology and Health Care Management, Chang Gung University of Science & Technology, Taoyuan 333324, Taiwan

**Keywords:** emergency room, workplace violence, resilience, intention to leave

## Abstract

(1) Background: Healthcare workplace violence has been a focused issue in the whole world. The rate of the occurrence is pretty high in every country. The emergency room is a high risk and high frequency place for violence to occur. Under the medical service demands from people, it is quite easy to bring about conflicts. This leads to serious physical and mental harm to nurses. When suffering from physical and mental injuries, resilience is a protective factor away from negative influence. It is rare to explore and study how the nurses’ resilience ability, workplace violence and turnover intention are related. Thus, the aim of this study is to understand resilience as a mediator effect in emergency nurses toward the workplace violence. (2) Methods: A cross-sectional survey study was used to collect information from emergency room nurses of a medical center in northern Taiwan. There were 132 samples in total. Three research instruments were included as follows: Hospital Workplace Violence Prevention Questionnaire, Connor-Davidson Resilience Scale, and Turnover Intention Scale. Statistical analysis using *t*-test, ANOVA, Correlation, as well as Sobel test were used in this study. (3) Results: The results revealed that the average age was 29.5 ± 5.6. Almost 58% of nurses experienced workplace violence. Twelve percent of nurse had experienced physical violence and 53.8% had experienced mental violence. There was significant relationship between shift personnel and religious believers. To the people who suffered physical violence, there was a significant relationship between emergency room working years and the total working years. There was significant difference between those who had suffered mental violence and religious believers. Female nurses suffered mental violence to a much higher extent than male nurses. There was a significant relationship between nurses’ working years, the total working years, resilience, and turnover intention. Resilience was not the mediator for workplace violence toward turnover intention in this study. (4) Conclusions: The outcome of this study suggested that on an individual level, nurses can enhance self-protection and communication skills to decrease workplace violence. For emergency environment settings, designing a good working environment, visitors’ restriction, avoiding working alone, and enhancing supervising alarm system are recommended. As for hospital administrators, fitness for work and to set up a project team is necessary. These can be references in planning prevention on workplace violence and promoting quality of workplace and patient safety in the future.

## 1. Introduction

Workplace violence has been a much researched and serious issue in the whole world. Healthcare organizations have been high frequency sites and it is especially common in the emergency room [1,2,3,4]. Workplace violence toward nurses is twice higher than toward the doctors and the other medical staff. They suffered beating 2.26 times more than the others [5,6]. The International Labor Organization (ILO), World Health Organization (WHO), International Council of Nurses (ICN) and Public Services International (PSI), defined mutually, in 2002, the definition of workplace violence as, “Incidents where staff are abused, threatened or assaulted in circumstances related to their work, including commuting to and from work, involving an explicit or implicit challenge to their safety, well-being or health,“ including physical violence and psychological violence [5].

The survey according to the Occupational Health Safety Network (OHSN) in America in 2012–2015 showed that the rate of emergency room violence was 19.3% and the average annual growth rate was 23% [6]. The occurrence rates in the other countries were 56–75.8% [1,7,8,9,10]. The study indicated only 30% occurrence rate had been officially reported [8,11] and up to 80% remained unreported [12]. Besides, every country has different definition on workplace violence, which leads to different data. The causes of emergency room violence were analyzed from four dimensions: inner and external factors, environmental factors and organizational factors. For instance, violence was part of work; there was poor communication skills; the waiting time was too long; doctor–patient information was no equal; the supervisors did not value at all; it was too crowded and noisy; it lacked privacy and there was a shortage of manpower, etc. [1,8,9,13,14]. Eventually workplace violence would reduce nurses’ passion for the job and it would result in health organizations losing experienced nurses [3].

Resilience is people’s ability to recover after experiencing adversity [15,16,17]. At present, there is no universal definition, but there are some common characteristics, such as adaptation and adjustment, dynamic process and ordinary magic [18]. First, the person must be exposed to the severe adversity; secondly, he/she can still maintain a positive adaptation under adversity [19,20]. “Workplace adversity” relates to heavy workloads, inexperience, professional autonomy influenced by bullying and violence, organizational problems, and occupational safety issues. These problems may be considered as adversity to people [21]. Nurses with high resilience see adversity as a part of their life, but not a threat [22]. Resilience decreases the influence of mental health to the minimum for nurses who work in the critical care units. It helps them stay in the workplace but not to choose to leave [21,23].

Turnover may be understood as employees leaving the organization or profession voluntarily. Turnover intention means mental status before quitting the organization or profession. It indicates that the employees are not satisfied with their jobs, coming about the ideas to quit, looking for the opportunities to work, evaluating and comparing with other opportunities to work. This situation takes shape before the actual turnover action happens. According to a study of Finnish labors on job disamenities, job satisfaction, and intentions to quit, variables such as education, discrimination, no promotion, heavy mentally, neglect, uncertainty, and harm had significant effects on switch intentions [24]. Choi and Lee [25] specified that 95.5% of nurses had experienced different workplace violence in the past one year. Not only were they exposed to the physical threat, but they were also experiencing slight to severe job burnout. There is a direct relationship between nurses’ experience of workplace violence and physical mental symptoms. The working pressure certainly impacts working satisfaction, increases turnover rate and the cost of the hospitals. In addition, it decreases their occupational life quality and healthcare quality and furthermore it becomes the factor of their turnover [26,27].

For more than 15 years of work in the emergency room, the researchers found out some of the nurses left the workplace after having experienced violence, and others were talking about leaving the current job. For those who stayed in the emergency room, they faced workplace violence either positively or ignored it. This phenomenon drew our attention to the relationship between workplace violence and intention to leave and why they still stay in the emergency room. The literature review indicated that healthcare professionals’ turnover intentions were mostly job burnout, emotional exhaustion, work pressure, and work satisfaction. Although there were researchers who surveyed the relationship between healthcare professionals’ resilience and turnover intention, only a few researchers explored the related studies about healthcare professionals’ workplace violence, resilience, and turnover intention. Therefore, the purpose of this study was to explore the effects of resilience as a mediator in emergency room violence toward nurses’ intention to leave.

## 2. Materials and Methods

### 2.1. Research Design and Samples

A cross-sectional survey study was used to collect information on emergency room nurses of a medical center in northern Taiwan. There were a total of 171 registered nurses in the emergency room and all of them were recruited in this study. Fifteen subjects refused to participate in the survey and 24 questionnaires were not filled out completely, leaving 132 valid questionnaires. The response rate was 84.6%. Inclusion criteria included: nurses at the emergency room worked over three months and were willing to participate.

### 2.2. Research Instruments

#### 2.2.1. Workplace Violence Instruments

The scale “Workplace Violence Instruments” was conducted by Institute of Labor, Occupational Safety and Health, Ministry of Labor, Taiwan, in 2014. There were five parts in the questionnaire, including basic data, physical violence, psychological violence (verbal abuse, bullied/mobbed and threats), policies and measurements, and the opinions toward workplace violence (open-ended question: three most important strategies to decrease violence in working environment.).

#### 2.2.2. Connor-Davidson Resilience Scale

The original scale was developed by Connor and Davidson [28] and it was divided into five levels and 25 questions, and the sequence of the levels were: (1) personal competence, high standards, and tenacity; (2) trust in one’s instincts, tolerance of negative affect, and strengthening effects of stress; (3) positive acceptance of change and secure relationship; (4) control; and (5) spiritual influences to evaluate the participants‘ ability to successfully respond to the pressure in the past one month. The present study used a 10-item CD-RISC Scale extracted by Campbell-Sills and Stein [29] from the original CD-RISC Scale. It is a Likert 5-point Scale and it respectively divides into Never (0), Occasionally (1), Sometimes (2), Often (3), and Always (4) from the lowest score 0 to the highest score 40. The higher the scores are, the higher the resilience is. The internal consistency of Cronbach’s α is 0.95 in the present study.

#### 2.2.3. Intention to Leave Scale

The present study used the Professor Huang Kai-Yi’s Chinese scale translated from Mobley’s [25] Turnover Intention Scale. It was mainly to measure the intention of employees’ leaving jobs. There were four questions including (1) the idea to leave, (2) the motivation to look for another job, (3) the degree of influence of the external job’s opportunity, and (4) the willingness to leave the current job. The questionnaire adapted 7-point to be the scoring method, from the lowest score 4 to the highest 28. The higher the scores are, the higher the turnover intention is. The internal consistency of Cronbach’s α is 0.72 in the present study.

### 2.3. Statistical Method

The Statistical Package Software of SPSS (IBM, Armonk, NY, USA) for Window 18.0 version was used for statistical analysis. Research data counted the obvious level and set *p* value < 0.05 as the standard to judge statistical meaning. Data analysis method included descriptive statistics, Independent Sample *t*-test, One Way Analysis of Variance, Chi-Square Test, Pearson Product-moment Correlation Coefficient, and Sobel Test Path Analysis.

## 3. Results

### 3.1. The Current Situation between the Participants’ Demographic Characteristics and Workplace Violence

A total of 132 participants were recruited in this study. The average age was 29.5 ± 5.6; the average overtime per week was 0.2 h; the average working years in the emergency room was 6.8 ± 5.1; the average total working years was 7.5 ± 5.6; shift personnel was 81.8%; nurses with in-service training on violence prevention was 62.9%; religious believers was 34.1%. In the last year, 57.6% of nurses suffered workplace violence. Twelve percent of nurses experienced physical violence and 53.8% of nurses suffered mental violence. The average resilience scored was 26.48 ± 6.59 and the average turnover intention score was 15.43 ± 4.76 (see Table 1).

Sixteen (12.1%) participants had suffered physical violence within one year. The main reasons were the patients’ condition changed too fast; the patients’ waiting time was too long; there was bad communication between nursing staff and patients. When the participants encountered the violence, they usually decided to stop the abusers to cease their behaviors and others would ask security guards for help. For mental violence, there were 71 (53.8%) participants who suffered verbal insult. The reasons were due to the patients’ waiting being too long, bad communication, and a crowded environment. Those who suffered bullying were 12 (9.1%). The bullying mainly came from their colleagues or supervisors and the bullying victims usually did not take any action about it. The personnel suffered threat was 20.5%. The probable causes were the patients’ waiting being too long, bad communication, crowded environment, terrible traffic flow, and that the patients’ condition changed too fast. When the participants faced mental violence, they usually decided to stop the abusers to cease their behaviors and the others would ask security guards for help. The reasons why they did not report mainly were they thought it was useless; it was not important; or they were afraid of negative results; or they did not know to whom they should report.

### 3.2. The Comparison of Difference among the Participants’ Workplace Violence, Resilience and Turnover Intention

There was a significant difference among workplace violence nurses in shifts and religion. Nurses with shift rotation (X^2^ = 4.840, *p* < 0.05 *) and religion (X^2^ = 5.121, *p* < 0.05 *) had higher frequency of workplace violence. There was significant difference among those who experienced physical violence with working years in the emergency room and the total working years. Nurses with less working years in the emergency room (t = −2.05, *p* < 0.05 *) and less total working years (t = −1.991, *p* < 0.05 *) tended to have experienced physical violence. As far as for mental violence, there was significant difference in religion. Nurses who were religious suffered mental violence more (X^2^ = 8.243, *p* < 0.01 **) (see Table 2).

### 3.3. The Related Comparison among the Participants’ Workplace Violence, Resilience, and Turnover Intention

There was a positive correlation between the working years in the emergency room and age (r = 0.934, *p* < 0.01 **). The total working years and age had a positive correlation (r = 0.959, *p* < 0.01 **). Satisfaction of dealing with physical violence had a negative correlation with age (r = −0.308, *p* < 0.01 **), working years in the emergency room (r = −0.305, *p* < 0.01 **) and total working years (r = −0.311, *p* < 0.01 **) (see Table 3).

### 3.4. The Intermediary Effects of Resilience

To test whether resilience was the mediator effect for the workplace violence and turnover intention, the Sobel test path was used according to Baron and Kenny’s [30] method. Using resilience as the mediator variable, the results of path analysis for those who had suffered workplace violence and turnover intention were shown as below:

There was not any significant relationship between nurses who suffered workplace violence and turnover intention (c = 0.091, *p* > 0.05).

No significant relationship showed between nurses who suffered workplace violence and resilience (a = −0.100, *p* > 0.05).

There was not a significant relationship between workplace violence victims’ resilience and turnover intention (b = −0.097, *p* > 0.05).

Resilience was not the mediator between nurses who suffered workplace violence and turnover intention (*p* > 0.05) (see Table 4).

## 4. Discussion

### 4.1. Workplace Violence

In the present study, 57.6% of participants had suffered workplace violence within one year, which was similar to previous research [1,7,8,9,10]. From patients’ point of view, the main reasons were long waiting time, bad communication, crowded environment, and their personal feelings. To nurses, the reasons were that the conditions of the patients changed too fast, the staff′s skill problems, and nurses’ attitude toward patients. Di Martino [5] pointed out the related reasons in the medical workplace violence could divide into work design and workload, such as ambiguity, overload work, lack of control, death dealing; the interpersonal relationship at work, such as conflict with the other staff; relationship with the patients and their family members, such as, insufficient preparation, insufficient mood dealing, and the demands of the patients and their family members; work organization and work management, for example, lack of the staffs’ support, employees’ turnover, the management echelon, and the supervisors’ difficulties, lack of resource and personnel shortage; the nursing techniques, such as treatment and nursing problems; personal, such as professional knowledge and skills [10,31,32,33]. It is recommended that nurse managers re-evaluate work assignment, re-enforce in-house education, and provide enough resources for emergency room nurses.

During the study, senior nurses always arranged with junior nurses, which increased workload for senior nurses. Meanwhile junior nurses suffered workplace violence during night shift and working alone. These reasons led to the results of this study that workplace violence was significantly correlated to shift rotation. There was a significant relationship between the workplace violence occurrence and shifts because the workload, time management, and the degree of busy would relatively enhance [22,32]. Tahghigit et al. [34] indicated that those who fitted the shifts well would be more highly pressure resistant. In this study, the religion was correlated with workplace violence. Nurses with religion tended to suffer workplace violence. It was probably based on Taiwanese′s culture that most Taiwanese would search for the help of gods [35,36] in the first place when facing troubles. It is recommended that junior nurses should avoid working at night shift and alone as possible. Nursing managers should pay more attention to nurses who are religious so as to early detect whether they are the victims of workplace violence.

Resilience in this study was not a mediator for intention to leave, which was different from other studies that showed that resilience was the mediator [34,37]. This might be due to that the institutions continue to hold in-service training regarding identification of workplace violence and nurses can identify from the first place and see it as part of the work.

### 4.2. Physical Violence

There were 12.1% of nurses suffered physical violence in the past one year as compared to 19–28.6% [38,39] in Taiwan and 10–42.7% in other countries [11,31,40,41], which is low. One of the reasons was when suffering violence they usually tolerated and chose not to report [3,40]. Verbal insult usually was the sign of physical violence on abusers [42]. It is recommended the security guards stationed around the clock. When it came to verbal insult, the nurse can inform the security guard immediately and it may stop the physical conflicts before the abusers took his act of violence. In addition, there was usually low notification rate in physical violence [11] with high turnover rate for nurses [42], which might give support the reason why there was lower physical violence or the victims might have left in the study. Hence, there was correlation between physical violence and intention to leave, and yet no correlation between mental violence and intention to leave.

There was significant correlation between working years in the emergency room and physical violence, which was similar to the past studies [10,32,43,44]. Studies revealed that workplace violence is related to age, inexperience, past experience of violence, and lack of communication skills [31,32]. The senior nurses were good at simplifying the events and then dealing with them [45]. They were good at distinguishing situations and having better negotiation skills [31]. Experience can be taught through learning to distinguish the signs before violence, communication skills, and negotiation skills.

### 4.3. Mental Violence

This study revealed that 53.8% of nurses suffered mental violence. Among these, 53.8% was verbal insult, 9.1% was bullying, and 20.5% was threats, which were similar to the previous research [25,31,32,38,39,40,46,47]. Verbal violence was five times as much as physical violence [7]. Workplace violence was mainly mental violence [10,25,38] or non-physical violence [31]. Verbal violence was the highest [10,25,38]. Abusers with verbal insults mainly came from the patients’ family members or the patients themselves. Bullying came mostly from colleagues and supervisors, results which are similar to previous studies [25,36,41,48,49]. The main causes of violence were the delay of examination and treatment, or dissatisfaction with quality of care [31,32,45,50]. Violence between the colleagues might be due to the responsibility to protect patients [10]. Nurses were concerned about it being useless to report or were afraid of the negative consequence, which was similar to the past research [49]. Yet, the silent culture inside the organizations would impact the results of the report [43]. Moreover, some nurses complained of not knowing whom they should report to. Therefore, knowledge and systems of prevention of violence should be disseminated to all employees in the institution.

## 5. Conclusions

There were 57.6% participants who suffered workplace violence in the past one year, while 12.1% suffered physical violence and 53.8% suffered mental violence. Abusers mainly came from patients (32.6%) and patients’ relatives and friends (35.6%). Violence occurred mainly because patients’ conditions changed too fast, long waiting times, bad communication, crowded environment, and bad traffic flow. When facing violence, the victims would stop the abusers at the first moment and secondly, they would ask the security guards for help. There was a relationship between workplace violence and shifts, religious belief, working years in the emergency room, and total working years. Resilience was not the intermediary variable for workplace violence and turnover intention.

The present study offered suggestions that could be divided into three parts, including individuals, environment, and organization. For nurses, it is important to enhance self-protect, the ability to communicate and coordinate with the patients and their family members as well as to use positive communication skills. For the environment factor, emergency administrators should reinforce the safety control at the main entrance, mark precisely the area and the route, and avoid the environment to work alone, and increase surveillance cameras and alarm systems. As for the organization factor, hospital administrators should adjust the work, allot the work properly, set up task forces toward case management modes, offer conformably the case’s mental health rebuilding system, present medical and legal assistance and improve police–civilian cooperation. The study concerned sensitive issues, so we adapted an anonymous questionnaire. Still, we could not exclude the participants not giving exact answers because they tried to fend off the sensitive issues. Therefore, an Internet questionnaire and APP application software are strongly advised in the future. In addition, the reason for low physical violence rate may be due to victims leaving jobs. It is recommended that the violence issue be included in nurses’ exit interviews, for a better understanding in future studies.

There are still a high percentage of unreported workplace violence around the world [6,10,13]. Compared to other countries′ workplace violence rate, 56–75.8%, this research showed a lower rate of 57.6%. This study provides workplace violence prevention strategies as references for other healthcare organizations internationally.

## 6. Limitation

Resilience in this study was not a mediator for the intention to leave, which were non-expected results. As for demographic data such as age, total working years, and emergency room working years, there was no correlation with intention to leave as well. This might be due to the institution continues to hold in-service training regarding identification of workplace violence, and nurses can identify it in the first place and see it as part of the work.

This study focused on the relationship between workplace violence and the intention to leave a job. Demographic data such as sex, age, education, marital status, working years, shift, training, and religion were included and yet some other non-occupational factors such as behaviors and emotional balance were not included. These variables will be considered in the future study.

## Figures and Tables

**Table 1 healthcare-09-00507-t001:** Current Analysis of Participants’ Demographic Data, Resilience, and Turnover Intention (*n* = 132).

Variable	M ± SD	N	%
Sex			
men		12	9.1
women		120	90.9
Age	29.5 ± 5.6		
Education			
college		12	9.1
university		120	90.9
Marital status			
single		93	70.5
married		39	29.5
Working hours per week	40.2 ± 0.08		
Working years/ER	6.8 ± 5.1		
Working years/total	7.5 ± 5.6		
Shift			
yes		108	81.8
no		24	18.2
Training			
Yes		83	62.9
no		49	37.1
Religion			
yes		45	34.1
no		87	65.9
Workplace Violence			
yes		76	57.6
no		56	42.4
Physical Violence			
yes		16	12.1
no		116	87.9
Mental Violence			
yes		71	53.8
no		61	46.2
Resilience	26.48 ± 6.59		
low		43	32.6
middle		38	28.8
high		51	38.6
Intention to leave	15.43 ± 4.76		

**Table 2 healthcare-09-00507-t002:** The Comparison of Difference for the Participants Suffered Workplace Violence, Resilience, and Turnover Intention (*n* = 132).

	Workplace Violence			Physical Violence			Mental Violence		
M ± SD	Yes (*n* = 76)	No (*n* = 56)	*t*	Yes (*n* = 16)	No (*n* = 116)	*t*	Yes (*n* = 71)	No (*n* = 61)	*t*
Age	30.0 ± 5.7	29.0 ± 5.7	1.1	27.1 ± 4.8	29.8 ± 5.7	−1.8	30.3 ± 5.7	28.6 ± 5.6	1.0
Working years/ER	7.1 ± 5.0	6.5 ± 5.2	0.61	4.4 ± 3.5	7.2 ± 5.2	−2.1 *	7.4 ± 5.0	6.2 ± 5.1	0.69
Working years/total	7.8 ± 5.6	7.1 ± 5.6	0.7	4.9 ± 4.6	7.9 ± 5.7	−2.0 *	8.2 ± 5.6	6.8 ± 5.5	0.9
Resilience	25.9 ± 7.1	27.3 ± 5.8	1.1	24.5 ± 7.9	26.8 ± 6.3	1.3	26.1 ± 7.0	27.0 ± 6.1	0.8
Intention to leave	15.8 ± 4.8	14.9 ± 4.7	−1.0	13.9 ± 4.0	15.7 ± 4.8	1.4	15.9 ± 4.9	14.9 ± 4.6	−1.3
	**Yes (%)**	**No (%)**	***X*** **^2^**	**Yes (%)**	**No (%)**	***X*** **^2^**	**Yes (%)**	**No (%)**	***X*** **^2^**
Sex			0.31						0.1
Men	50.0	50.0		8.3	91.7		50.0	50.0	
Women	58.3	41.7		12.5	87.5		54.2	45.8	
Education			0.31						0.1
College	50.0	50.0					50.0	50.0	
University	58.3	41.7					54.2	45.8	
Marital status			0.9						0.1
Single	60.2	39.8		15.1	84.9		54.8	45.2	
Married	51.3	48.7		5.1	94.9		51.3	48.7	
Shift			4.8 *						3.1
Yes	62.0	38.0		13.0	87.0		57.4	42.6	
No	37.5	62.5		8.3	91.7		37.5	62.5	
Religion			5.1 *			0.1			8.2 **
Yes	71.1	28.9		13.3	86.7		71.1	28.9	
No	50.6	49.4		12.1	88.5		44.8	55.2	
Training			0.006			0.269			0.017
Yes	48	35		11	72		45	38	
No	28	21		5	44		26	23	

* *p* < 0.05, ** *p* < 0.01.

**Table 3 healthcare-09-00507-t003:** Related Comparison on Participants’ Demographic Data, Resilience, and Intention to leave (*n* = 76).

Variable	1	2	3	4	5	6	7
1 Age	1						
2 Working years/ER	0.934 **	1					
3 Working years/total	0.959 **	0.945 **	1				
4 Satisfaction of physical violence	−0.308 **	−0.305 **	−0.311 **	1			
5 Satisfaction of Mental Violence	0.068	0.109	0.061	−0.193	1		
6 Resilience	0.232 *	0.218	0.218	−0.055	−0.106	1	
7 Intention to leave	0.079	0.023	0.097	−0.276 *	−0.169	−0.159	1

* *p* < 0.05, ** *p* < 0.01.

**Table 4 healthcare-09-00507-t004:** Regression Analysis Chart with resilience as the mediation variable test (*n* = 132).

	IV	M	DV
	Workplace Violence	Resilience	Intention to leave
Step1			c = 0.091, *p* = 0.297
Step2			a = −0.100, *p* = 0.254
Step3			b = −0.097, *p* = 0.272
Step4			*p* = 0.427

## Data Availability

The datasets used and/or analysed during the current study are available from the corresponding author upon reasonable request.

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
