# Peer review of "The Effects of Emergency Room Violence toward Nurse′s Intention to Leave—Resilience as a Mediator"

_healthcare, 2021, doi:10.3390/healthcare9050507_

Round 1

Reviewer 1 Report

Comments

1.  The contribution to the (international) literature should be stated.
2.  The empirical context of the paper should be motivated in the revised introduction.
3.  Was non-response random or not?
4.  Do the data contain (survey) weights or not?
5.  The paper should acknowledge that there are earlier empirical studies on the effect of job characteristics on employee turnover using linked survey and register data (https://doi.org/10.1111/j.1468-232X.2008.00546.x). 
6.  The paper does not consider the heterogeneity in the estimated effects. The relationships between the variables of interest can differ significantly, e.g., by age/gender. The relatively small size (N=132) limits the analysis by subgroups.
7.  The concluding section of the paper state practical policy conclusions that stem from the results that are presented in the paper.

Author Response

Thank you for your precious comments. We have explained point-by-point according to your comments.

Reviewer 2 Report

First of all, I’d like to give my congratulations to the authors for the manuscript and for giving me the opportunity to read it. Secondly, I have some concerns as outlined below.

  • More justification is needed about the hypothesis and above the conceptualization of the variables in the study. Specially about the mediation role of resilience.
  • Please, revise more current bibliography.
  • Please, include the hypotheses of the study and the theoretical model in which the hypotheses are based.
  • More info about the sample and the procedure will be welcomed.
  • In page 5 the sentence ‘there was significative difference in religion’ need to be more specific. What is means?
  • What about the correlation matrix about violence, resilience and turnover intention? How the non-significant correlations could be explained?
  • Since the correlation matrix shows non-significant relationships among the main variables in the study, maybe it has non-sense to test the mediation analyses.

I hope that these previous comments are interesting for authors.

Author Response

(The authors gave the same response as above.)

Reviewer 3 Report

The research presented is quite interesting, focusing on a pertinent andtransversal to several countries. The authors centered their study onfactors of work, but it was never considered factors outside of work, associated to personal factors that may have had a direct influence on behavior and emotional balance of the professionals. There is no reference throughout the text to support what was previously mentioned, nor was it considered by the authors other very important variables associated with physical and emotional stress of the professionals, such as the number of continuous hours of daily work performed by health professionals, complementary training, health status of the professional, number of familiar elements, among others.Thus, in my opinion, the study could consider other variables that related to personal as well as work variables, that could influence the performance of the professional in situations of greaterstress and exposure to different types of violence. The results could actually be different, including other variables.The question is, were more elements related to professionals collected?What the average number of elements simultaneously in episodes of urgency?Are the identified hospitals, central hospitals?

Author Response

(The authors gave the same response as above.)

Round 2

Reviewer 2 Report

Dear authors, thanks for answering my questions. I think that the manuscript has been improved a lot. The only thing is that the non-correlations as well as the non-expected results of resilience should be well documented in the limitations section.

Best regards

Author Response

limitation section was added in page9 ,line324~335
